

# Transcriptomic, and metabolic profiling reveals adaptive mechanisms of *Auricularia heimuer* to temperature stress

Chenhong Nie[1,2,3], Shiyan Wei[2], Shengjin Wu[2], Liangliang Qi[2], Jing Feng[3] and Xiaoguo Wang[1,2]

[1] School of Resources, Environment and Chemistry, Chuxiong Normal University, Chuxiong, China
[2] Microbiology Research Institute, Guangxi Academy of Agriculture Sciences, Nanning, China
[3] School of Marine Sciences and Biotechnology, Guangxi Minzu University, Nanning, China

Corresponding authors
Jing Feng,
fengjing0414141@foxmail.com
Xiaoguo Wang,
wangxiaoguo2005@163.com

## ABSTRACT

Temperature significantly influences the growth and development of edible mushrooms, including the popular *Auricularia heimuer*. Despite its economic importance, the molecular mechanisms that enable *A. heimuer* to withstand prolonged temperature stress are poorly characterized. Here, we performed a comprehensive morphologic, transcriptomic, and metabolic analysis of *A. heimuer* mycelium exposed to different temperatures over a long period of time. Low temperatures (LT) suppressed mycelial growth, while high temperatures (HT) promoted it. Extremely high temperatures (EHT) were highly detrimental, not only inhibiting growth but also potentially leading to mycelial mortality. The production of reactive oxygen species (ROS) and the activities of antioxidant enzymes such as superoxide dismutase (SOD) and catalase (CAT) were significantly altered by temperature. Transcriptomic profiling identified 1,024, 778, and 4,636 differentially expressed genes (DEGs) in LT, HT, and EHT, respectively, compared to normal temperature (NT). The response to LT was found to involve the regulation of protein synthesis and transport. Notably, HT and NT shared the highest degree of similarity, indicating that these two conditions represent a moderate temperature range that places less stress on the mycelium. In contrast, exposure to EHT resulted in the upregulation of genes related to ribosomal biogenesis, suggesting that *A. heimuer* may increase protein synthesis in response to heat stress. Furthermore, many genes related to carbohydrate metabolism were downregulated under EHT. Enzymatic assays further confirmed that thermal stress profoundly affects the synthesis of metabolic byproducts and the activities of key glycolytic enzymes, suggesting a restructured metabolic landscape under stressful conditions. In summary, our comprehensive analysis of the *A. heimuer* mycelial transcriptomic and enzymatic responses to sustained temperature fluctuations provides valuable insights into the molecular basis of thermotolerance. This work lays the foundation for future breeding efforts aimed at improving the resilience of cultivated *A. heimuer* and can serve as the basis for similar initiatives in other fungal species.

## INTRODUCTION

Black wood ear (*Auricularia heimuer*), a popular edible and medicinal mushroom, has been cultivated for millennia in China (*Sun et al., 2022*). *A. heimuer* is rich in polysaccharides, melanin, and essential mineral elements (*Pak et al., 2021*). The polysaccharides isolated from *A. heimuer* have been found to exhibit antioxidant, immunomodulatory, and anticancer properties (*Nguyen et al., 2012*). Unlike other mushrooms, *A. heimuer* contains melanin, which is associated with antioxidant, anti-biofilm, and hepatoprotective effects (*Bin et al., 2012*; *Zhao et al., 2015*; *Zou, Zhao & Hu, 2015*; *Hou et al., 2019*). In addition, a high nutritional value and low fat content have made *A. heimuer* a sought-after ingredient with increasing market demand (*Yang et al., 2022*). Fruiting bodies constitute the primary agronomic commodity, but their yield stability is intricately linked to the mycelium's ability to maintain metabolic homeostasis and mitigate oxidative damage during temperature fluctuations, which constitutes a prerequisite for transitioning from vegetative growth to reproductive development (*Hu et al., 2023*).

Temperature can significantly impact the growth and development of edible mushrooms (*Sakamoto, 2018*; *Hu et al., 2023*). Extremely high or low temperatures can inhibit mycelial expansion (*Qiu et al., 2018*; *Yan et al., 2020*; *Yue et al., 2024*), alter physiological and biochemical profiles (*Yan et al., 2019*), disrupt growth cycles (*Martínez-Soto & Ruiz-Herrera, 2017*), and reduce disease resistance (*Yang et al., 2023*). Furthermore, exposure to extreme temperatures can trigger overproduction of reactive oxygen species (ROS), resulting in oxidative damage to critical molecules such as DNA, lipids, and proteins (*Hou et al., 2020*). Luckily, fungi have evolved a number of functional proteins to counteract temperature-related stress. For example, *Pleurotus ostreatus* upregulates the production of superoxide dismutase (SOD), catalase (CAT), and ascorbate peroxidase (APX) in response to high temperatures (*Hu et al., 2023*). Similarly, *Stropharia rugosoannulata* increases the expression of antioxidant enzymes to combat low-temperature stress (*Hao et al., 2022*). In addition, heat shock proteins (HSPs) and other associated proteins are crucial for managing cellular stress and preventing protein misfolding, thereby improving resilience to extreme temperatures (*Chen, Feder & Kang, 2018*; *Abu Bakar, Karsani & Alias, 2020*). However, little is known regarding the temperature sensitivity of *A. heimuer*. Understanding the impact of temperature stress on *A. heimuer* will aid in optimizing cultivation management and improving yield and quality.

The majority of research has focused primarily on the effects of heat shock on fungal mycelia. While these studies provide valuable insights, they have tended to overlook the complexity associated with prolonged temperature adaptation. For example, *Pleurotus tuoliensis* mycelia exposed to 32 °C and 36 °C for 96 h exhibited upregulated biosynthesis of HSPs and ergosterol, both of which are crucial for adaptation to heat stress (*Chen et al., 2023a*). Proteomic analysis of heat-stressed *P. ostreatus* exposed to 40 °C for 48 h revealed the involvement of MAPK signaling, antioxidant defense, HSP production, and the glycolysis pathway in thermotolerance (*Zou et al., 2018*). *Hypsizygus marmoreus* exhibited upregulated CAT, SOD, peroxidase (POD), and trehalose biosynthesis in response to heat stress at 37 °C (*Xu, Guo & Yu, 2020*). Transcriptomic analysis of *Ganoderma lucidum*

mycelia exposed to 42 °C for 2 h highlighted 176 differentially expressed genes (DEGs) involved in carbohydrate metabolism and other biological pathways (*Tan et al., 2018*). Unfortunately, each of these studies focused only on short-term stress and may not have fully captured the dynamic responses of fungal mycelia to prolonged temperature stress.

In this study, we performed a comprehensive transcriptomic analysis of *A. heimuer* mycelia under prolonged temperature stress. Specifically, we sought to identify the molecular and physiological adaptations that enable *A. heimuer* to withstand temperature stress. We characterized transcriptome-wide changes in gene expression through transcriptomic sequencing, providing a sensitive, high-throughput, and broad-range view of the mycelial response. In addition, we examined the ROS and carbohydrate contents, as well as the activities of key enzymes, to obtain a holistic understanding of the metabolic mechanisms underlying mycelial growth and stress resistance in *A. heimuer*. This research improves our fundamental understanding of fungal thermotolerance and provides a genetic foundation for developing improved cultivation strategies to increase productivity.

## MATERIALS & METHODS

### Fungal materials and temperature treatment

*A. heimuer* strain 'CGMCC1536' was used for all assays and is deposited in the Chinese General Microbiological Culture Collection Center (CGMCC). The strain showed excellent performance in regional trials, with a short mycelium growth cycle, good thermotolerance, and favorable fruiting body morphology. *A. heimuer* mycelia were cultured on potato dextrose agar (PDA) plates under the following conditions: low temperature (15 °C, LT), normal temperature (25 °C, NT, as control), high temperature (35 °C, HT), and extremely high temperature (45 °C, EHT). To ensure that all mycelia started at the same initial viability level, the inocula for each treatment were prepared from the same starting plate (full of mycelium) and then cultured in the dark at 28 °C for 3 days. On the 4th day following inoculation, the four groups of experimental plates were transferred to incubators set at 15 °C, 25 °C, 35 °C, and 45 °C. All plates were incubated in the dark for five days for subsequent analyses.

### Mycelial growth rate and fresh weight measurements

Radial growth of the mycelium was measured in millimeters. Incubation was terminated when the rapidly growing mycelium was observed to completely colonize the culture dish. To ensure the collection of intact biomass, a sterile cellophane membrane was pre-layered onto PDA prior to inoculation. The biomass-covered membrane was peeled off, and the mycelium was gently scraped using a sterile spatula. Fresh weight (g) was recorded immediately with an analytical balance. Each treatment consisted of five independent replicates.

### Reagents and assay kits

Standardized assay kits (Suzhou Comin Biotechnology Co., Ltd., Jiangsu, China) were used to measure ROS content, enzymatic activity, and carbohydrate metabolism. All protocols were strictly followed according to the manufacturer's instructions.

## ROS content and antioxidant enzyme activity assays

The contents of hydrogen peroxide ($H_2O_2$), the lipid peroxidation marker malondialdehyde (MDA), and the osmoregulatory amino acid proline (Pro), as well as the activities of SOD, CAT, POD, lignin peroxidase (Lip), and laccase, were evaluated. Details about the assay methods can be found in Table S1.

## Carbohydrate metabolism analysis

We analyzed the contents of sucrose, glucose, fructose, and fructose-1,6-diphosphate (FDP), all of which are essential substrates for energy production as well as metabolic intermediates. The activities of key enzymes related to the glycolytic pathway and sugar conversion were also evaluated, including invertase, hexokinase (HK), phosphofructokinase (PFK), and pyruvate kinase (PK). Taking into account the possible shift toward anaerobic metabolism under thermal stress, we also examined the content of lactic acid (LA), a common end product of anaerobic glycolysis, and the activity of lactate dehydrogenase (LDH), which catalyzes the interconversion of pyruvate and lactate.

## RNA extraction and transcriptomic sequencing

Total RNA was extracted from the mycelia using TRIzol® Reagent (Invitrogen, Waltham, MA, USA) according to the manufacturer's instructions. RNA quality was determined using a 5300 Bioanalyser (Agilent Technologies, Santa Clara, CA, USA) and quantified using an ND-2000 (NanoDrop Technologies, Waltham, MA, USA). RNA purification, reverse transcription, library preparation, and sequencing services were provided by Shanghai Majorbio Bio-pharm Biotechnology Co. Ltd. (Shanghai, China) following the protocol specified by the manufacturer. The RNA-seq transcriptome library was prepared following Illumina® Stranded mRNA Prep, Ligation workflow (San Diego, CA, USA) using 1 μg of total RNA. Quality assessment and trimming of the raw paired-end reads were performed using fastp (*Chen et al., 2018*) with standard settings. Subsequently, the clean reads were individually aligned to the reference genome using the HISAT2 aligner (*Kim, Langmead & Salzberg, 2015*). For each sample, the aligned reads were then consolidated using StringTie (*Pertea et al., 2015*) in a reference-guided assembly approach. The *A. heimuer* reference genome (GCA_002287115) was downloaded from the NCBI GenBank. Each experiment was conducted with three biological replicates and five technical replicates for transcriptomic analysis. Additionally, the statistical power of the experimental design (0.72) was calculated in RNASeqPower.

## Analysis of differentially expressed genes

Transcript expression levels were determined using the transcripts per million (TPM) metric. Gene expression levels were quantified using RSEM (*Li & Dewey, 2011*). Statistically significant DEGs were identified using DESeq2 (*Love, Huber & Anders, 2014*) according to the absolute log2-fold change ($|log2FC| \geq 1$) and false discovery rate (FDR) $\leq 0.05$ criteria. In addition, Gene Ontology (GO) and Kyoto Encyclopedia of Genes and Genomes (KEGG) functional enrichment analyses were performed using a Bonferroni-corrected *P*-value threshold of $\leq 0.05$ relative to the entire transcriptome. The GO enrichment and KEGG

pathway analyses were facilitated by GOatools (https://github.com/tanghaibao/GOatools) and Python scipy software (https://scipy.org/install/), respectively.

## RESULTS

### Effect of temperature on the growth rate and fresh weight of *A. heimuer* mycelia

Mycelial growth varied significantly with temperature (Fig. 1A). As temperature increased, the mycelial growth rate first increased and then decreased, peaking at 35 °C (HT) (Fig. 1B). Mycelial fresh weight exhibited a similar trend, with the highest fresh weight observed at 35 °C (HT) and the lowest at 45 °C (EHT) (Fig. 1C). Morphologically, the mycelia were white, dense, and robust when cultured at LT, NT, and HT. At EHT, the mycelia exhibited macrophenotypic regression and significantly inhibited growth. Overall, mycelial growth was suppressed at low temperatures, enhanced at moderate and high temperatures, and severely inhibited at extremely high temperatures. Notably, the effects of high temperatures were more dramatic than the effects of low temperatures in terms of mycelial growth.

### ROS dynamics of fungal mycelia across a range of growth temperatures

We sought to understand how temperature affects ROS dynamics in *A. heimuer* mycelia. Specifically, we evaluated the contents of specific ROS and the activities of antioxidant enzymes in mycelia exposed to different temperatures. The content of $H_2O_2$ decreased with increasing temperature, likely indicating decreased oxidative stress or the adjustment of antioxidant defenses (Fig. 2A). The activity of SOD, which neutralizes superoxide radicals, increased with increasing temperatures (Fig. 2B). Likewise, the activity of CAT, which neutralizes $H_2O_2$, increased with increasing temperatures (Fig. 2C). These results suggest that high temperatures increase the demand for ROS scavengers such as SOD and CAT. The content of MDA decreased with increasing temperatures (Fig. 2D). The activity of POD, which neutralizes peroxides, exhibited a bimodal pattern wherein activity decreased at the lowest and highest temperatures, peaking at HT (Fig. 2E). The Pro content decreased with increasing temperatures (Fig. 2F). Finally, the activities of Lip and laccase were highest at HT and EHT, respectively, likely due to the increased levels of oxidative stress at these temperatures (Figs. 2G–2H).

### Comprehensive transcriptomic analysis

A total of 90.07 GB of clean data were generated from all 12 samples, with a Q30 > 94.31% (Table S2). The data were deposited with the NCBI Sequence Read Archive (SRA) under accession number PRJNA1208126. Clean reads were mapped to the *A. heimuer* reference genome (GCA_002287115) with a mapping rate of 83.81%-85.05%. A total of 11,354 expressed genes were detected across all samples (Table S3). In general, the sequencing data obtained were of excellent quality and were therefore suitable for subsequent analyses.

### Differential gene expression in response to different temperatures

Principal component analysis (PCA) indicated that gene expression under LT, NT, HT, and EHT were well-clustered and clearly separated (Fig. 3A), implying a different response

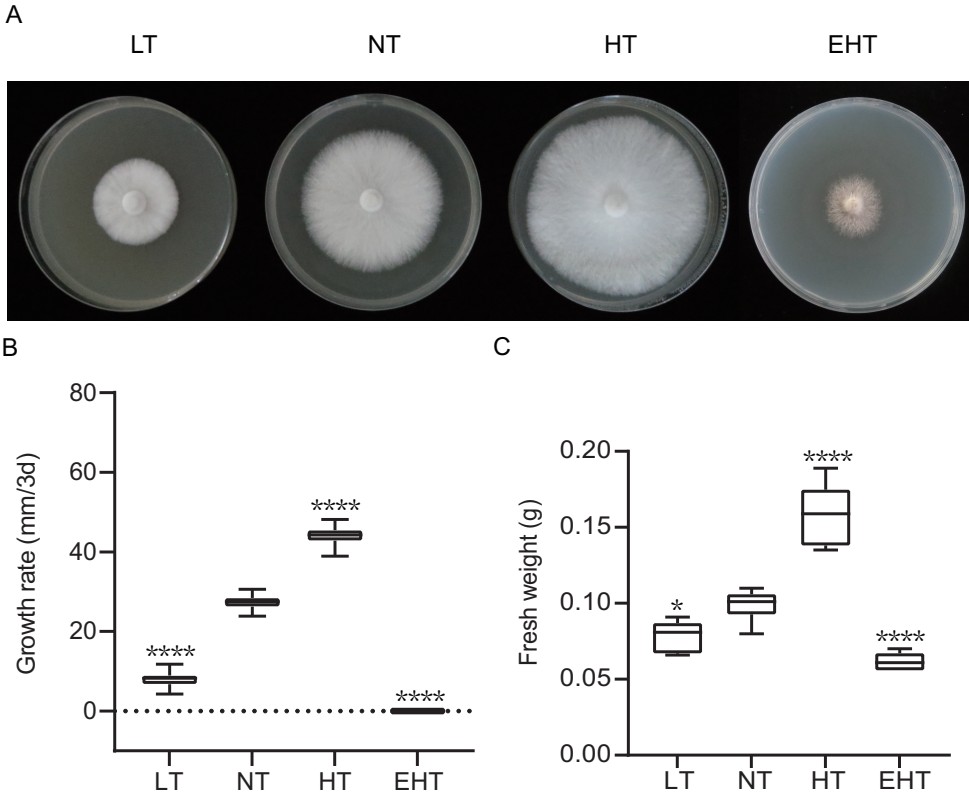

**Figure 1 The growth of *A. heimuer* mycelia at different temperatures.** (A) *A. heimuer* mycelia on potato dextrose agar (PDA) under different temperatures. (B) Growth rate and (C) fresh weight of *A. heimuer* mycelia on PDA under different temperatures. LT, low temperature, 15 °C; NT, normal temperature, 25 °C; HT, high temperature, 35 °C; EHT, extremely high temperature, 45 °C. The data are presented as relative expression levels compared with the control group. ns indicates $p > 0.05$ (no statistical significance), $*p \leq 0.05$, $**p \leq 0.01$, $***p \leq 0.001$, $****p \leq 0.0001$.

to each level of temperature stress. A large number of DEGs were detected in LT (465 upregulated, 559 downregulated), HT (308 upregulated, 470 downregulated), and EHT (2,163 upregulated, 2,473 downregulated) relative to NT (Fig. 3B). Mycelia exposed to HT and NT shared the highest level of similarity, with each exhibiting minimal differential gene expression. In contrast, mycelia exposed to EHT exhibited the most significantly differential gene expression compared to NT (Fig. S4).

In a comparison of gene expression profiles, LT *vs.* NT shared 57 co-expressed DEGs with HT *vs.* NT and 491 co-expressed DEGs with EHT *vs.* NT. Furthermore, 366 co-expressed DEGs were observed between HT *vs.* NT and EHT *vs.* NT. A core set of 179 DEGs was shared among all three comparison groups, suggesting a common response to temperature fluctuations (Fig. 4). These shared DEGs were found to be involved in several important pathways related to carbohydrate metabolism. For example, many DEGs were involved in fructose and mannose metabolism, and these sugars not only serve as energy resources but also precursors in the synthesis of other molecules. Other DEGs were involved in starch and

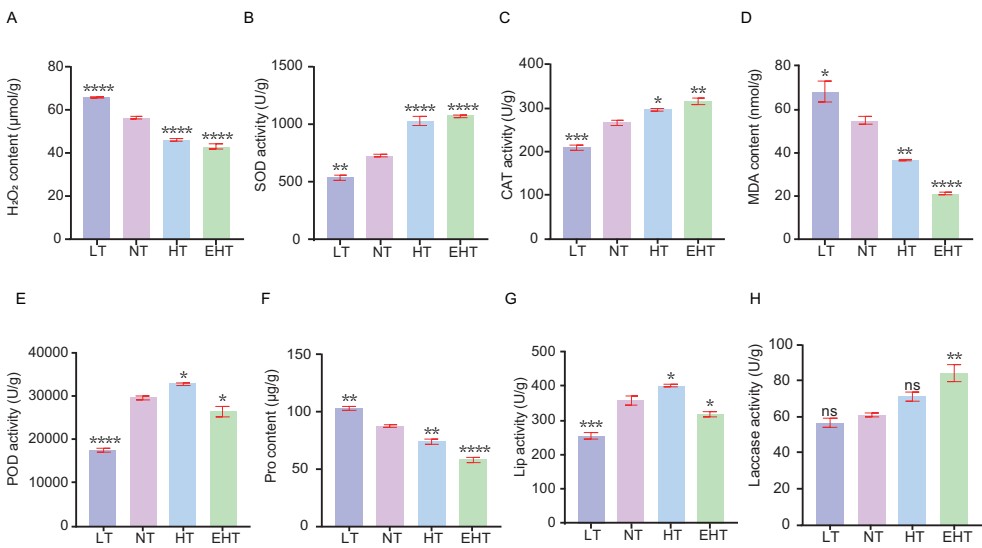

**Figure 2** **The contents of oxidative-stress related molecules and the activities of antioxidant enzymes in *A. heimuer* mycelia at different temperatures.** (A) Hydrogen peroxide ($H_2O_2$) content. (B) Superoxide dismutase (SOD) activity. (C) Catalase (CAT) activity. (D) Malondialdehyde (MDA) content. (E) Peroxidase (POD) activity. (F) Proline (Pro) content. (G) Lignin peroxidases (Lip) activity. (H) Laccase activity. LT, low temperature, 15 °C; NT, normal temperature, 25 °C; HT, high temperature, 35 °C; EHT, extremely high temperature, 45 °C. The data are presented as relative expression levels compared with the control group. ns indicates $p > 0.05$ (no statistical significance), $*p \leq 0.05$, $**p \leq 0.01$, $***p \leq 0.001$, $****p \leq 0.0001$.

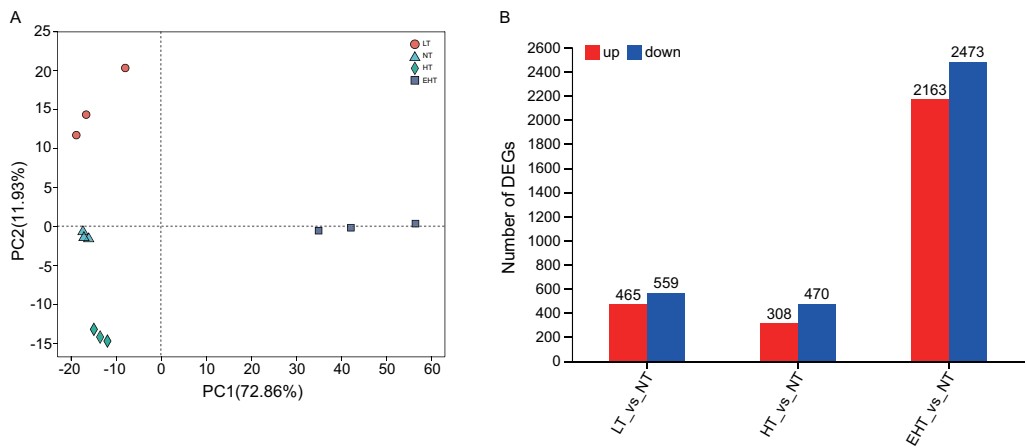

**Figure 3** **Differential gene expression in *A. heimuer* mycelia at different temperatures.** (A) Principal component analysis (PCA) of gene expression at different temperatures. (B) The number of differentially expressed genes (DEGs) ($p$-adjusted $\leq 0.05$, $|log2(fold change)| \geq 1$) under different temperatures. LT, low temperature, 15 °C; NT, normal temperature, 25 °C; HT, high temperature, 35 °C; EHT, extremely high temperature, 45 °C.

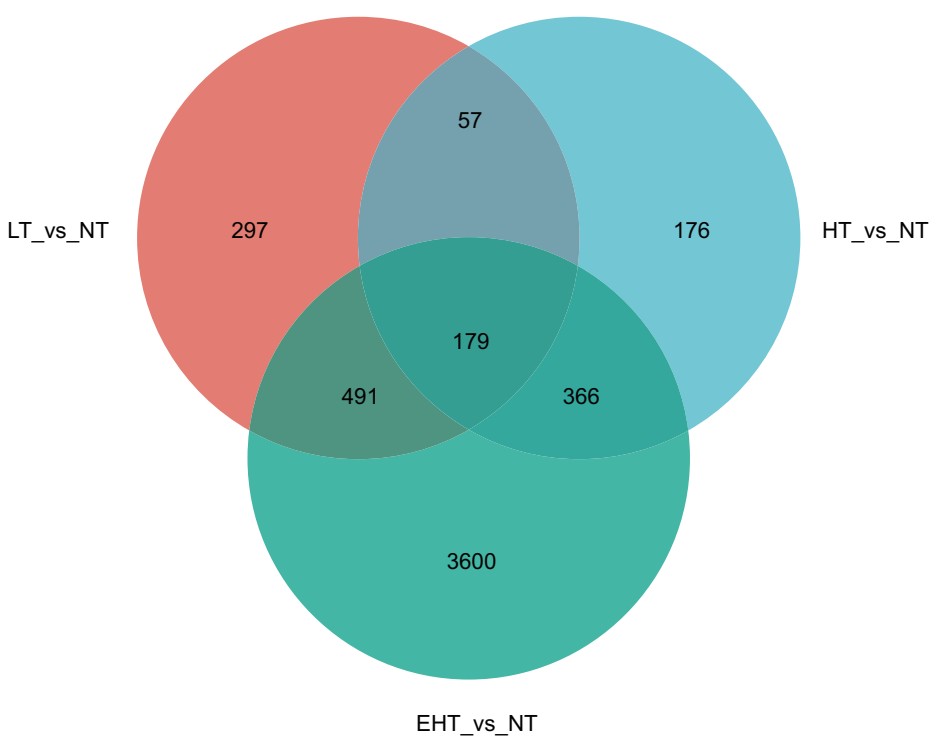

**Figure 4** **Venn diagram of differentially expressed genes (DEGs) in *A. heimuer* mycelia at different temperatures.** LT, low temperature, 15 °C; NT, normal temperature, 25 °C; HT, high temperature, 35 °C; EHT, extremely high temperature, 45 °C.

sucrose metabolism, which supports energy production and other cellular functions. The pentose phosphate pathway, also known as the oxidative phase of the pentose phosphate pathway, plays a role in pentose metabolism and NADPH formation. Ascorbate and aldarate metabolism is associated with antioxidant synthesis and cellular redox homeostasis. Finally, the amino sugar and nucleotide sugar metabolism pathway is essential for the synthesis of proteins, nucleotides, and other molecules that play critical roles in cellular structure and function. That the shared DEGs were enriched in these particular pathways suggests that sugar metabolism is central to the fungal response to temperature stress. Therefore, regulation of these metabolic pathways may be an adaptive strategy to overcome metabolic challenges associated with fluctuating or extreme temperatures (Table S4).

## Functional annotation and enrichment analysis

To investigate the biological processes associated with each DEG, we identified the most significant GO terms in all three comparison groups ($p$-adjusted $\leq 0.05$). All GO terms fell into the following three categories: molecular functions, cellular components, or biological processes. The 20 most frequently annotated terms were quite similar among LT *vs.* NT, HT *vs.* NT, and EHT *vs.* NT (Figs. 5A–5C; Table S5). All three term categories were among the significant annotations, and included catalytic activity, binding, membrane, cell part, metabolic process, and cellular process. To further explore pathways affecting mycelial

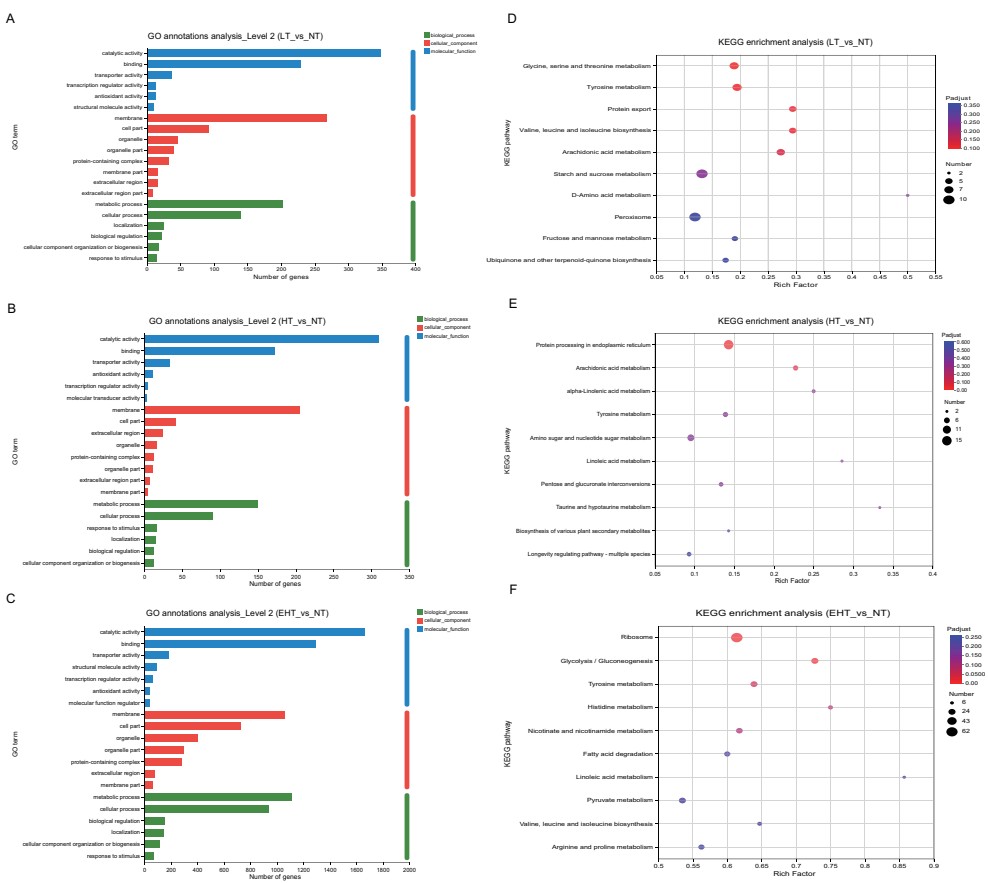

**Figure 5   Functional analysis of differentially expressed genes (DEGs) in *A. heimuer* mycelia at different temperatures.** (A–C) Gene Ontology (GO) annotation. (D–F) Kyoto Encyclopedia of Genes and Genomes (KEGG) pathway enrichment analysis. LT, low temperature, 15 °C; NT, normal temperature, 25 °C; HT, high temperature, 35 °C; EHT, extremely high temperature, 45 °C.

development at different culture temperatures, we used rich factors and adjusted *p*-values as criteria to identify the ten most enriched pathways in each comparison group. The rich factor is the ratio of the number of DEGs annotated with a pathway term to the total number of genes annotated with that pathway term. A higher rich factor combined with a lower adjusted *p*-value signals significant enrichment of the corresponding KEGG pathway. The expression levels and fold changes of DEGs in the pathways mentioned below are presented in Tables S6–S8.

The significant enrichment of DEGs in key metabolic pathways suggests metabolic reprogramming, which may be a strategic adaptation to low-temperature conditions. In LT *vs.* NT, the upregulated genes were involved in glycine, serine, and threonine metabolism (map00260); tyrosine metabolism (map00350); the biosynthesis of valine, leucine, and isoleucine (map00290); and arachidonic acid metabolism (map00590) (Fig. 5D). These pathways may play a critical role in maintaining membrane fluidity, protecting macromolecules, and modulating the synthesis of compatible solutes to maintain cellular

homeostasis and ensure survival in cold conditions. Protein export (map03060) was also enriched, and the active secretion of proteins may also therefore serve as an adaptation to cold stress. The rich factors of each of these pathways were 0.19, 0.19, 0.29, 0.29, and 0.27, respectively, highlighting their importance to cold tolerance. Such extensive metabolic reprogramming highlights the complexity of the fungal response to low temperatures and provides valuable insights into the molecular strategies which may be employed to improve cold tolerance in agriculture and horticulture.

Pathways associated with HT *vs.* NT highlighted the importance of protein homeostasis and energy metabolism in the fungal response to high-temperature stress. Protein processing in the endoplasmic reticulum (map04141) and arachidonic acid metabolism (map00590) were significantly enriched. With rich factors of 0.14 and 0.22, respectively, these pathways highlight the biological strategies used to maintain cellular integrity and metabolic efficiency under thermal stress (Fig. 5E). Specifically, these pathways are likely crucial for the proper folding, modification, and transport of proteins, as well as for the synthesis and degradation of arachidonic acids, which are essential for membrane structure and function. As indicated by their rich factors, these pathways were significantly overrepresented among DEGs, suggesting a coordinated metabolic response to the challenges posed by high temperatures. These results provide potential targets for genetic manipulation to improve thermotolerance in agricultural and horticultural applications as well as valuable insights into the molecular basis of the high temperature response.

Several key biological pathways were enriched in EHT *vs.* NT. The ribosome (map03010) is essential for protein synthesis, and its significant enrichment suggests upregulation of protein production in response to extreme heat stress. Likewise, the significant enrichment of the glycolysis/gluconeogenesis pathway (map00010), which is involved in the utilization of carbohydrates for energy production, suggests a metabolic shift to meet increased energy demands (Fig. 5F). Tyrosine metabolism (map00350) and histidine metabolism (map00340) were also significantly enriched, with rich factors of 0.64 and 0.75, respectively. These results suggest that protein synthesis may serve a critical function in heat stress adaptation. Such high rich factors (0.61, 0.73, 0.64, and 0.75) indicate the significant overrepresentation of these pathways among the DEGs, highlighting their importance to the extreme heat stress response (Fig. 5F).

Together, these results suggest that fungi undergo metabolic reprogramming under extreme temperature conditions, prioritizing protein synthesis, energy production, and amino acid metabolism. Such metabolic reprogramming is likely a crucial adaptive strategy to thermal stress, allowing fungi to survive under high temperatures. These results provide a deeper understanding of the molecular mechanisms underlying the fungal response to extreme temperatures and may aid in the development of strategies to improve thermotolerance in agricultural and horticultural applications.

## Differential gene expression in response to temperature stress

Significant shifts in gene expression were observed in LT *vs.* NT, affecting both metabolic pathways and protein export mechanisms. In particular, the tryptophan synthase gene

[EC:4.2.1.20], which participates in glycine, serine, and threonine metabolism, was found to be upregulated (Fig. S1A). The upregulation of this gene suggests a likely increase in the synthesis of L-tryptophan, an essential amino acid for protein synthesis and precursor to various biologically-active compounds. Conversely, the expression levels of genes encoding enzymes involved serine and threonine degradation were downregulated in response to low temperatures. These include L-serine/L-threonine ammonia lyase [EC:4.3.1.17 4.3.1.19], D-serine ammonia lyase [EC:4.3.1.18], threonine dehydratase [EC:4.3.1.19], D-amino-acid oxidase [EC:1.4.3.3], and sarcosine oxidase/L-pipecolate oxidase [EC:1.5.3.1 1.5.3.7] (Fig. S1A). Downregulation of these enzymes would be expected to result in reduced pyruvate synthesis, an important intermediate in the citric acid cycle, indicating a metabolic shift in response to cold stress. Regarding protein export, upregulation of the SRP9, SRP14, SRP72, SPCS2, and SPCS3 genes (Fig. S1B) suggests improved regulation of protein synthesis and transport in response to cold stress. The upregulation of these genes is likely crucial for maintaining cellular function and integrity under low temperature conditions. Notably, the CAT and SOD2 genes were downregulated, while the SOD1 gene was upregulated, in the peroxisome pathway (Fig. 6). These changes may be indicative of a readjustment of the cellular antioxidant response under low temperature conditions. Interestingly, the EPHX2 gene exhibited variable expression, with one gene (gene08410) downregulated and two genes (gene08416; gene05774) upregulated (Fig. 6). EPHX2 is involved in the inactivation of biologically-active lipid aldehydes and variable expression of this enzyme may reflect a complex regulatory response to low temperature conditions, potentially involving different cellular compartments or response stages. In summary, the observed changes in gene expression under low temperature conditions, including the upregulation of tryptophan synthase and protein export-related genes and the downregulation of certain catabolic enzymes and antioxidant defense genes, are reflective of a coordinated response to alter metabolism and protein processing. These results provide valuable insights into cellular-level adaptations to low temperature conditions.

Cells produce HSPs in response to stressful conditions, particularly high ambient temperatures. HSPs act as molecular chaperones capable of folding, assembling, and transporting other proteins, and can break down misfolded proteins to prevent cellular damage. The expression levels of the Hsp40, Hsp70, and Hsp90 genes within the endoplasmic reticulum protein processing pathway were downregulated in HT *vs.* NT. Conversely, the expression levels of nine out of ten Hsp20 genes were upregulated (Fig. S2). Hsp20 proteins, also known as small heat shock proteins, stabilize protein structures and prevent protein aggregation. Their upregulation suggests a cellular response to safeguard proteins against thermal denaturation. Furthermore, the membrane-associated gene phospholipase B1 [EC:3.1.1.4 3.1.1.5], associated with arachidonic acid metabolism, was upregulated (Fig. 7). The EPHX2 gene [EC:3.3.2.10] was also upregulated (Fig. 7), likely representing a protective mechanism to counteract the increased production of potentially harmful lipid aldehydes arising from arachidonic acid metabolism under stressful conditions.

Ribosome-related genes were significantly upregulated in EHT *vs.* NT (Fig. 8A), indicative of the increased production of ribosomal proteins. Increased ribosome biogenesis

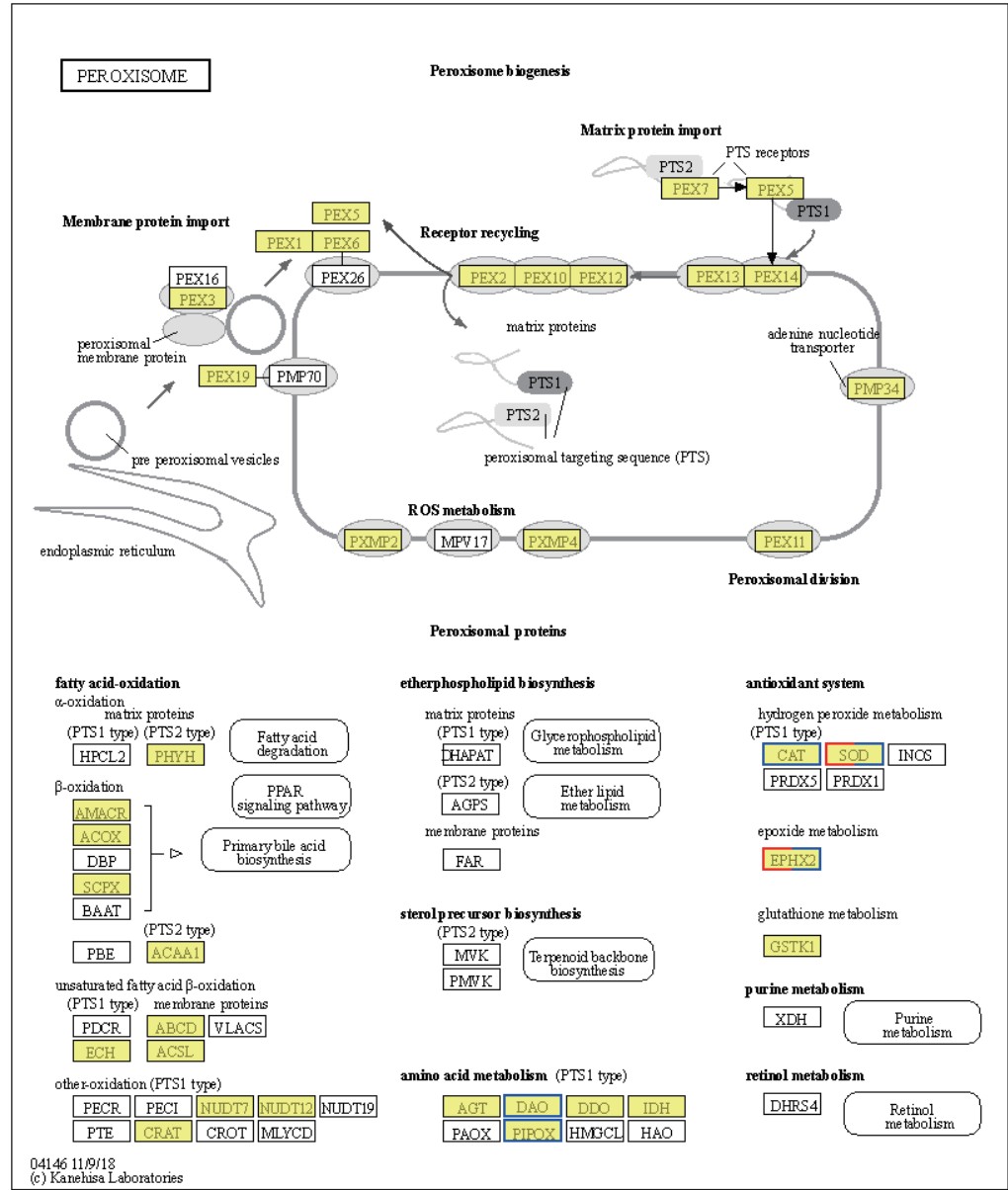

**Figure 6  Kyoto Encyclopedia of Genes and Genomes (KEGG) pathway annotations of differentially expressed genes (DEGs) in LT *vs*. NT.** Annotated map of peroxisome. Yellow highlights indicate known genes, red frames indicate upregulated genes, blue frames indicate downregulated genes, and half-red-half-blue frames indicate genes that are both upregulated and downregulated. LT, low temperature, 15 °C; NT, normal temperature, 25 °C.

may be a cellular response to the increased need to synthesize stress-responsive proteins and other molecules under high-temperature stress (Fig. S3A). In contrast, genes involved in the generation of pyruvate in the glycolysis/gluconeogenesis pathway were predominantly downregulated (Fig. 8). This downregulation may indicate a reduction in the rate of glycolysis and energy production. On the other hand, expression of the PDHB gene was

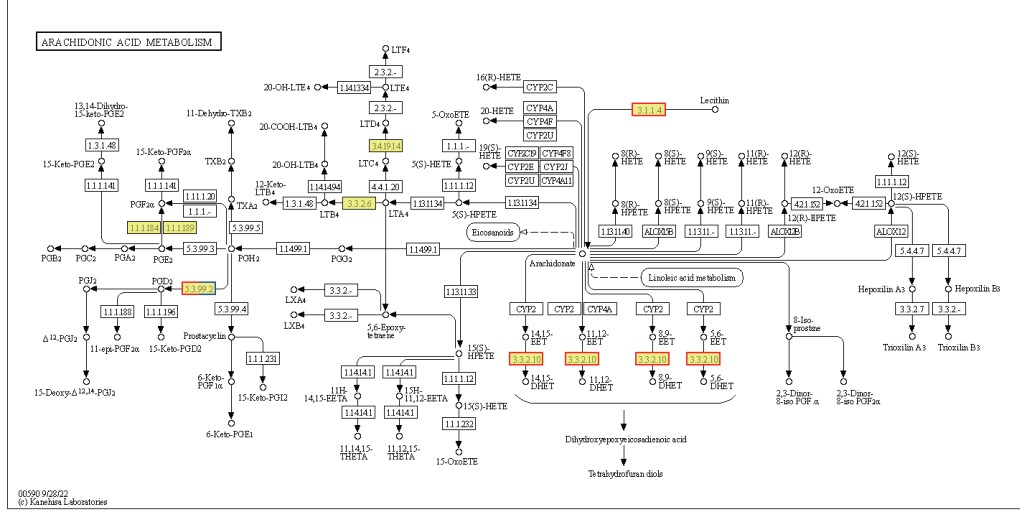

**Figure 7** **Kyoto Encyclopedia of Genes and Genomes (KEGG) pathway annotations of differentially expressed genes (DEGs) in HT *vs.* NT.** Annotated map of arachidonic acid metabolism. Yellow highlights indicate known genes, red frames indicate upregulated genes, blue frames indicate downregulated genes, and half-red-half-blue frames indicate genes that are both upregulated and downregulated. NT, normal temperature, 25 °C; HT, high temperature, 35 °C.

upregulated. PDHB is part of the pyruvate dehydrogenase complex that converts pyruvate to acetyl-CoA for the citric acid cycle, and its upregulation may be an adaptation to meet the increased energy demands associated with the heat stress response. Furthermore, genes involved in the synthesis of L-histidine were mostly upregulated (Fig. S3B). L-histidine is an essential amino acid that plays various roles, including as a precursor for the synthesis of histamine, which is involved in the immune response. The upregulation of these genes may be reflective of an increased requirement for L-histidine under high-temperature conditions, possibly for the synthesis of molecules that support cellular stress response or for the maintenance of thermo-sensitive cellular functions.

## Temperature-dependent metabolic shifts in *A. heimuer* mycelium

We evaluated the influence of thermal stress on the metabolic profile of *A. heimuer* mycelia by analyzing the contents of key metabolites and the activities of key enzymes involved in carbohydrate metabolism. The contents of sucrose (Fig. 9A), glucose (Fig. 9B), and fructose (Fig. 9C), and the activities of invertase (Fig. 9G), hexokinase (Fig. 9H), and phosphofructokinase (Fig. 9I), tended to increase with increasing temperature. However, the contents and activities of each of these molecules decreased at EHT, suggesting that HT in this study served as the optimal metabolic temperature, with both higher and lower temperatures resulting in stress. Likewise, the contents of fructose-1,6-diphosphate (Fig. 9D) and pyruvic acid (Fig. 9E), as well as the activity of pyruvate kinase (Fig. 9J), followed a similar pattern. The peak activity at HT suggests the temperature-dependent regulation of glycolytic enzymes, which are crucial for energy production and the formation of metabolic intermediates. In contrast, the content of lactic acid (Fig. 9F) showed a biphasic

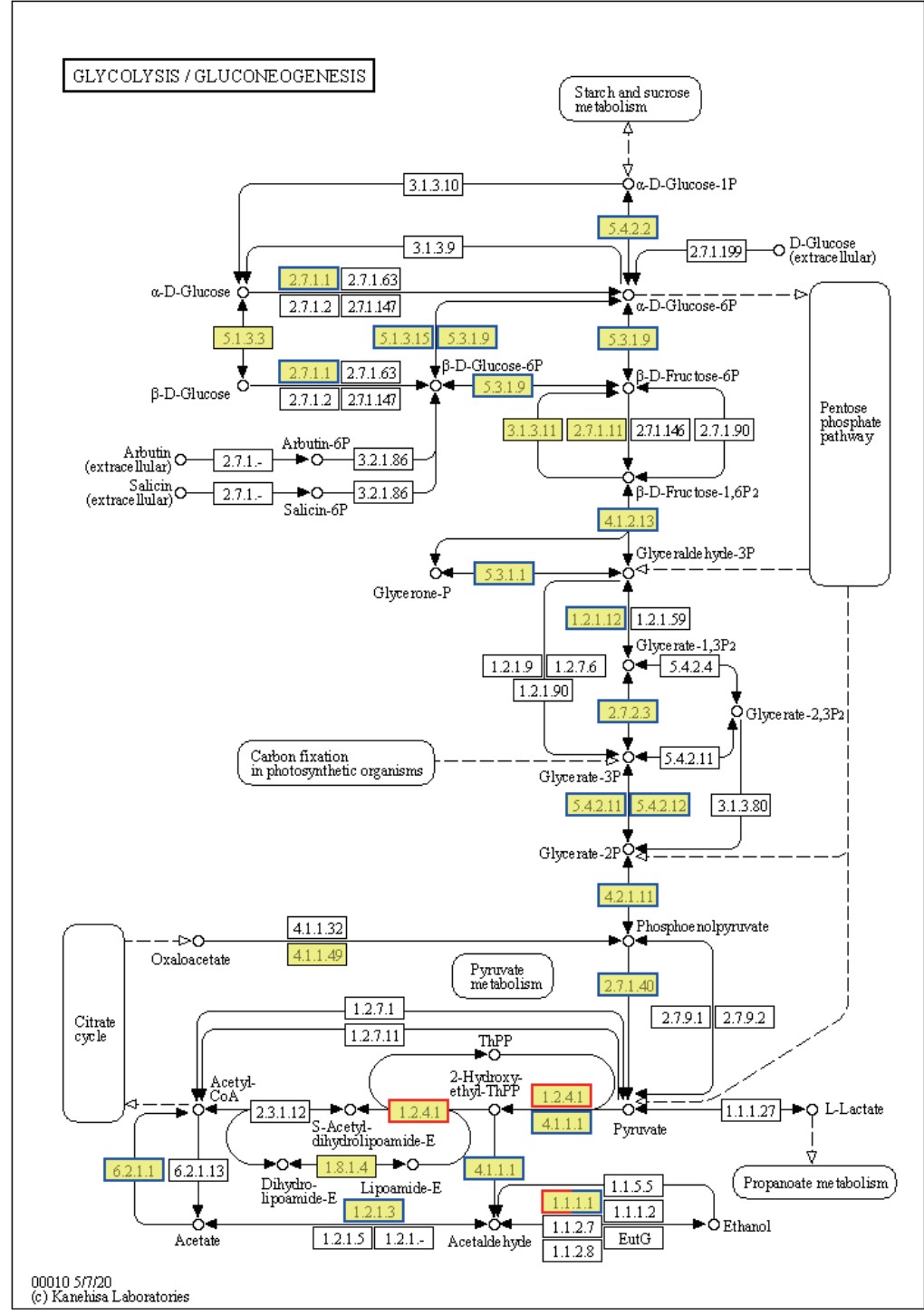

**Figure 8 Kyoto Encyclopedia of Genes and Genomes (KEGG) pathway annotations of differentially expressed genes (DEGs) in EHT *vs*. NT.** Annotated map of glycolysis/gluconeogenesis. Yellow highlights indicate known genes, red frames indicate upregulated genes, blue frames indicate downregulated genes, and half-red-half-blue frames indicate genes that are both upregulated and downregulated. NT, normal temperature, 25 °C; EHT, extremely high temperature, 45 °C.

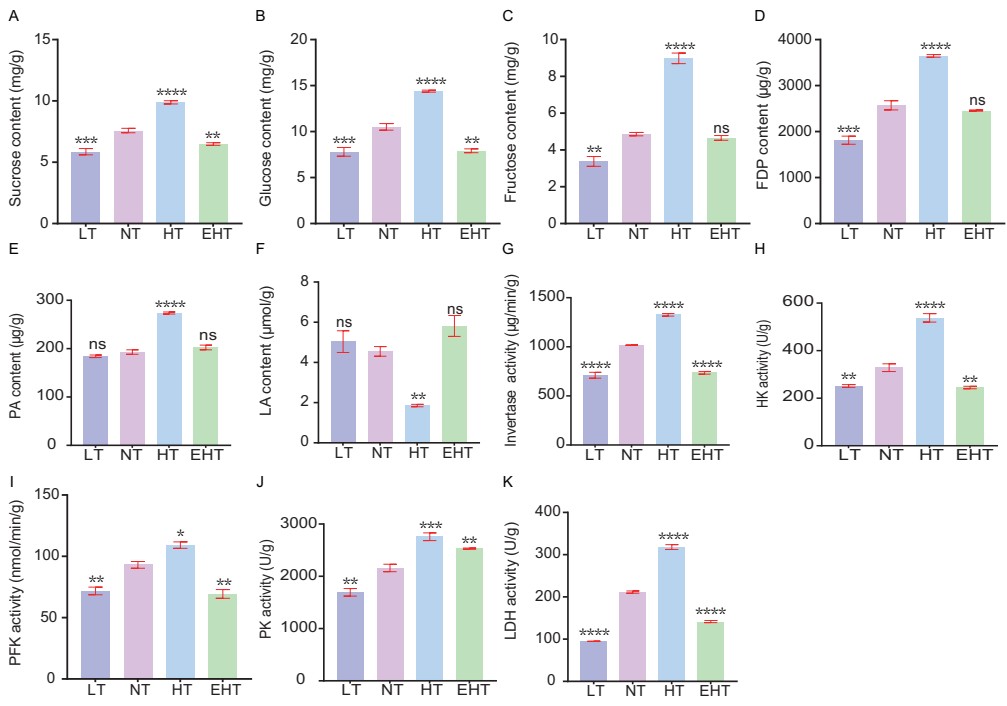

**Figure 9** **The contents of carbohydrate-metabolism related molecules and the activities of key enzymes in *A. heimuer* mycelia at different temperatures.** (A) Sucrose content. (B) Glucose content. (C) Fructose content. (D) Fructose-1, 6-diphosphate (FDP) content. (E) Pyruvicacid (PA) content. (F) Lactic acid (LA) content. (G) Invertase activity. (H) Hexokinase (HK) activity. (I) Phosphofructokinase (PFK) activity (J) Pyruvate kinase (PK) activity. (K) Lactate dehydrogenase (LDH) activity. LT, low temperature, 15 °C; NT, normal temperature, 25 °C; HT, high temperature, 35 °C; EHT, extremely high temperature, 45 °C. The data are presented as relative expression levels compared with the control group. ns indicates $p > 0.05$ (no statistical significance), $*p \leq 0.05$, $**p \leq 0.01$, $***p \leq 0.001$, $****p \leq 0.0001$.

response, initially decreasing with increasing temperatures but peaking at EHT. This may indicate a shift to anaerobic metabolism or a protective response to heat stress. The activity of lactate dehydrogenase (Fig. 9K), which catalyzes the conversion of pyruvate to lactate, exhibited the opposite trend: increasing with increasing temperature, but decreasing at EHT. Together, these results indicate that anaerobic respiration may become increasingly important at high temperatures.

Notably, the results indicate that high temperatures more strongly alter metabolic processes than low temperatures in *A. heimuer* mycelia. Increasing metabolism at higher temperatures may be an adaptive response to maintain metabolic homeostasis. However, the metabolic decline observed at EHT suggests that *A. heimuer*'s metabolic machinery is sensitive to extreme temperatures, resulting in reduced growth and possible cell damage. These observations highlight the importance of temperature in regulating metabolism as well as the potential of thermal stress to influence the growth and metabolic efficiency of *A. heimuer* mycelia. Understanding these responses is crucial for optimizing growth conditions and developing strategies to mitigate the effects of temperature fluctuations on fungal metabolism.

## DISCUSSION

### Enhanced protein synthesis and transport at low temperatures

In *A. heimuer*, mycelial growth is significantly inhibited by low temperatures, resulting in the upregulation of genes involved in protein synthesis and transport (*Fang et al., 2024*). The increased expression of small subunit processome components such as SRP9, SRP72, and SRP14 likely underlies a mechanism for improving the regulation of protein synthesis and transport in response to cold temperatures. This adaptation is crucial for maintaining cellular functions and improving physiological adaptability under stressful conditions (*Jiao et al., 2024*). In addition to the transcriptional regulation of protein synthesis and transport, fungal adaptation to low temperatures often involves the remodeling of membrane lipid composition to maintain fluidity and functionality. In response to cold stress, many fungi increase the proportion of unsaturated fatty acids in phospholipid membranes to prevent excessive rigidity and preserve membrane integrity (*Los & Murata, 2004*). This lipid phase adaptation likely operates synergistically with the upregulated SRP-mediated protein trafficking, as identified in our study (*Stjepanovic et al., 2011*). The coordinated enhancement of both membrane plasticity and protein synthesis efficiency may represent a dual strategy for sustaining cellular homeostasis and stress resilience in the face of low temperature-induced growth inhibition.

The upregulation of tryptophan synthetase in cold-stressed mycelia may reflect an increased demand for tryptophan to support essential growth and metabolic processes (*Chen et al., 2023b*). Tryptophan is a critical amino acid required for protein synthesis and various metabolic functions (*Jia et al., 2024*). The tryptophan biosynthesis pathway is complex and involves multiple enzymes and regulatory steps. While the upregulation of tryptophan synthetase could indicate an attempt to increase tryptophan production, the overall impact on tryptophan levels depends on the interplay of various factors within this pathway. The upregulation of tryptophan synthetase may be part of a broader metabolic response to maintain cellular homeostasis under cold stress (*Zhou et al., 2025*). This type of regulation may be particularly important when environmental temperatures affect metabolic rates and amino acid availability (*Cossins & Lee, 1985*).

### Antioxidant defense and metabolic adjustment at high temperatures

Here, we observed the downregulated expression of high molecular weight HSPs in response to high temperatures, perhaps due to reduced need. On the other hand, the expression of low molecular weight HSPs was upregulated. This metabolic adaptation may be aimed at achieving optimal growth conditions under high temperature stress. In general, HSPs play a critical role in cellular proteostasis and the heat shock response. However, different HSPs have different roles and regulatory mechanisms under thermal stress. For example, larger HSPs such as Hsp70 and Hsp90 act as chaperones to assist in protein folding and prevent protein aggregation, while smaller HSPs such as Hsp27 regulate cellular processes under stressful conditions (*Lang et al., 2021*). The differential expression of HSPs in response to temperature changes has been observed in other fungi. For example, in *Ganoderma lingzhi*, heat stress stimulates the binding of phosphatidic acid to mTOR, resulting in its activation. In turn, the activated mTOR regulates the core form of SREBP, suggesting that a complex

regulatory network controlling mTOR biosynthesis modulates ganoderic acid synthesis in response to heat stress (*Liu et al., 2024*). The expression and activity of HSPs may be closely linked to temperature-responsive metabolic pathways and cellular processes, allowing mycelia to adapt to high temperature environments by modulating the expression of HSPs to maintain cellular integrity and support growth and development.

The high temperature-induced upregulation of PLB and EPHX2 may improve membrane fluidity and permeability, thereby making the exchange of materials between the inside and outside of the cell more efficient. These results are consistent with findings that treatments resulting in increased phospholipid bilayer (PLB) fluidity can affect the permeability of the membrane to reactive species (*Yusupov et al., 2017*). In addition, the released fatty acids can serve as signaling molecules and regulate the cellular response to changing environmental conditions. This has been observed in research on lipid signaling in plants, highlighting the role of lipids in adaptation to stressful environmental conditions (*Oubohssaine, Hnini & Rabeh, 2024*).

The mycelial growth rate was highest at HT, indicating that this temperature may be within the optimal range for the 'CGMCC1536' strain. Enzymatic assays further supported this, showing increased activities of key glycolytic enzymes like hexokinase (HK), phosphofructokinase (PFK), and pyruvate kinase (PK) at HT. The higher enzyme activities suggest enhanced glycolysis and energy production. Additionally, higher levels of reducing sugars and efficient conversion of glucose to fructose-1,6-diphosphate (FDP) at HT indicate more active carbohydrate metabolism, consistent with the observed growth enhancement.

## Ribosomal protein synthesis and metabolic reorganization at extremely high temperatures

The ribosome, as a central player in protein synthesis, may play a key role in cellular adaptation to high temperatures (*Farewell & Neidhardt, 1998*; *Wang et al., 2024*). Here, we observed that exposure to EHT resulted in the upregulation of ribosomal protein genes, indicating a potential shift in gene expression patterns as a response to heat stress. By modulating the expression of ribosomal genes, cells might attempt to maintain critical cellular functions and stability under heat stress (*Volkov, Panchuk & Schöffl, 2003*). This finding offers insights into the potential molecular mechanisms of adaptation to heat stress and underscores the importance of ribosomal gene regulation in this process. The upregulation of ribosomal genes may primarily reflect transcriptional responses, while their actual translational activity could be modulated by additional regulatory layers. These possibilities warrant further validation through translatomic or proteomic analyses.

Temperature changes can significantly affect metabolic activity. Histidine is an essential component of cell membranes and signaling systems and plays a critical role in maintaining membrane stability and facilitating intracellular signaling (*Schwentner et al., 2019*). Its similarity to cell membrane lipids and ability to form stable complexes with signaling molecules ensure the maintenance of membrane stability and signal transduction. Although mycelial growth was observed to be minimal at extremely high temperatures, increased

histidine synthesis may represent a heat stress adaptation involving the protection of membrane stability and the maintenance of intracellular signaling.

The downregulation of genes encoding enzymes in the glycolysis pathway, such as pyruvate kinase, indicates a weakened ability to convert sugar into energy. Under thermal stress, *A. heimuer* mycelia may be subjected to pressure resulting in altered energy requirements and usage efficiency (*Fang et al., 2024*). The upregulation of PDHB and the downregulation of PDC may reflect the reconfiguration of metabolic pathways in response to heat stress, and the likely prioritization of alternative energy production pathways or heat stress responses. Pyruvate dehydrogenase, which converts pyruvate to acetyl-CoA, may be more active at high temperatures to support energy production or the formation of important metabolites.

The significant upregulation of ribosomal proteins and the metabolic shifts observed at extremely high temperatures indicate a cellular strategy to enhance protein synthesis and energy production. These results demonstrate that *A. heimuer* has evolved a proactive approach to combat the deleterious effects of heat stress, possibly by upregulating stress response proteins and cellular repair mechanisms.

## CONCLUSIONS

In this study, we comprehensively evaluated the mechanisms by which *A. heimuer* adapts to extreme temperatures by integrating phenotypic, transcriptomic, and metabolic analyses. Our results revealed that the *A. heimuer* strain exhibits a high growth rate at 35 °C, suggesting a broader optimal temperature range than previously reported. The numbers and functions of DEGs were found to vary with temperature. Notably, low temperatures did not cause irreversible damage to mycelial cells and the low temperature response primarily involved enhanced antioxidant defenses and upregulation of protein transport mechanisms. High temperatures more significantly affected mycelial growth, and the high temperature response involved the accumulation of HSPs, the regulation of fatty acid metabolism, and the production of antioxidant enzymes. These results contribute to a better understanding of fungal temperature tolerance mechanisms and provide new insights into optimizing the cultivation and breeding of *A. heimuer*.

### Funding

This research was supported by the Science and Technology Fund of Guangxi Academy of Agricultural Sciences (2021YT094, and 2025YP075), the Yunnan Province "Xingdian Talent Support Program" Young Top Talents Special Project (YN-WR-QNBJ-2020-104), the Modern Agriculture and Innovation of Agricultural Organization System of Guangxi (Nycytxgx-cxtd-2021-07-02), the China Agriculture Research system (CRAS20), the Key Research and Development Project of Guangxi (AB22035030) and the National Key Research and Development Project of China (2022YFD1100104). The funders had no role

in study design, data collection and analysis, decision to publish, or preparation of the manuscript.

## Grant Disclosures

The following grant information was disclosed by the authors:

Science and Technology Fund of Guangxi Academy of Agricultural Sciences: 2021YT094, 2025YP075.

Yunnan Province "Xingdian Talent Support Program" Young Top Talents Special Project: YN-WR-QNBJ-2020-104.

China Agriculture Research system: CRAS20.

Key Research and Development Project of Guangxi: AB22035030.

National Key Research and Development Project of China: 2022YFD1100104.

## Competing Interests

The authors declare there are no competing interests.

## Author Contributions

- Chenhong Nie conceived and designed the experiments, performed the experiments, analyzed the data, prepared figures and/or tables, authored or reviewed drafts of the article, and approved the final draft.
- Shiyan Wei conceived and designed the experiments, authored or reviewed drafts of the article, and approved the final draft.
- Shengjin Wu conceived and designed the experiments, authored or reviewed drafts of the article, and approved the final draft.
- Liangliang Qi conceived and designed the experiments, authored or reviewed drafts of the article, and approved the final draft.
- Jing Feng conceived and designed the experiments, analyzed the data, authored or reviewed drafts of the article, and approved the final draft.
- Xiaoguo Wang conceived and designed the experiments, performed the experiments, analyzed the data, prepared figures and/or tables, authored or reviewed drafts of the article, and approved the final draft.

## Data Availability

The data is available at NCBI SRA: PRJNA1208126.

## Supplemental Information

Supplemental information for this article can be found online at http://dx.doi.org/10.7717/peerj.19713#supplemental-information.

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
