# Peer review of "Transcriptomic, and metabolic profiling reveals adaptive mechanisms of Auricularia heimuer to temperature stress"

_PeerJ, doi:10.7717/peerj.19713_

## Round 0.1 · original submission · Major Revisions

Two experts assessed your manuscript and reported several observations that need attention. The most relevant ones include the lack of statistical analyses and controls. Moreover, additional experimentation is required to support some statements.

Reviewer 1 ·

Basic reporting

Correction
- Page 9 line 156: changes “di-erentially” to “differentially”
- Page 14 line 356: changes “(Fig. 8H)” to “(Fig. 9H)”
- Figure 1: Italicize A. heimuer.
- Figure 5: Gene Ontology (GO) annotation A) is sorted from the lowest to highest number of genes, unlike B) and C), which are sorted from highest to lowest. Please revise them to be sorted in the same way.
- Figure 6-8: What does it mean for both upregulated and downregulated in one gene for example, EPHX2 in Figure 6? In the text page 13 Line 313: the EPHX2 gene exhibited variable expression, with one case of downregulation and two cases of upregulation (Fig. S1 B). Is it the variation among three biological replicates? Add a clear definition.
- Figure 7: In text page 14 Line 332The EPHX2 gene was also upregulated (Fig. 7). I have not seen EPHX2. Label it on the picture. By this, researchers should ensure that the differentially expressed genes (DEGs) mentioned in the text are consistent with those presented in the figures.

Methods description
- Page 8 line 114 Specify how to separate mycelia from PDA to measure fresh weight. However, dried weight is more suitable for growth measurement.
- The researchers should perform statistically on the results in Fig 1, 2, and 9 to clarify the increase and decrease of growth rates, metabolic products or enzymatic activities.

Results
- Are there transcription factors that are differentially expressed in this experiment? Discuss about potential transcription factors involved in these responses.
- Researchers should add clustering and trend analysis of differentially expressed genes and mentioned fold-changed of expressed genes in text.

Experimental design

Title
- Consider removing 'Morphologic' from the title, as it represents only a minor aspect of the overall content of the article.
- If researchers would like to mention morphology changes, you should add morphology of mycelia at microscopic level and discuss about morphological changes.

There are 2 concerns about experimental design of this research.
- Actually, under high-temperature (HT) conditions, Auricularia heimuer mycelia grow better than under normal-temperature (NT) conditions. Could we then consider HT as the optimal condition for A. heimuer and, accordingly, reconsider renaming NT and low-temperature (LT) to low temperature (LT) and extremely low temperature (ELT), respectively? In my opinion, we should not refer to a condition in which the organism grows better as a stress condition — it may simply reflect its adapted optimal growth environment.

- At extremely high temperatures, does Auricularia heimuer survive? It showed no growth. Given the high number of differentially expressed genes—up to 3,000—under this condition compared to others, it raises the question of whether this is due to fluctuating gene expression during the dying process. (Some publications have mentioned this phenomenon.) Therefore, researchers should verify whether the mycelia remain viable after exposure to extreme high temperature (up to 45°C) for five days or check whether they formed survival structure such as sclerotium.

Validity of the findings

- The researchers should conduct some experiments to support the hypothesis. For example, on page 15, line 396, it is suggested that improved tryptophan synthesis may activate certain downstream cold-responsive metabolic pathways. To test this, an experiment could be designed to compare the cold adaptation of Auricularia heimuer grown in medium with and without added tryptophan under cold conditions. Such an experiment would enhance the credibility of the research.

Reviewer 2 ·

Basic reporting

1、line 5 "A. heimuer" in the abstract is not in italics. There is no space between the words "differenttemperatures"
2、Missing Key Literature: Discussions on low-temperature adaptation (lines 382-397) omit classic studies on fungal membrane lipid phase changes, weakening comparative analysis of cold tolerance mechanisms.
3、Disconnect Between Background and Objectives: The introduction highlights the "economic importance" of Auricularia heimuer but focuses solely on mycelial stages, ignoring temperature effects on fruiting body development—a critical agronomic trait.
4、Poor Visualization: Figures 2 and 9 omits statistical significance markers (e.g., asterisks, letters), preventing readers from assessing data reliability.

Experimental design

1、Treatments lasted only 5 days (Materials & Methods, line 113), whereas "long-term" in fungal physiology typically spans weeks/months. This short duration cannot mimic natural long-term stress.
2、10°C gap between HT (35°C) and EHT (45°C) misses transitional temperatures (37°C, 40°C), making it impossible to identify the critical threshold from adaptation to damage.
3、Controls and Replication Shortcomings Lack of Blank Controls: No uninoculated PDA plates were included as blanks, risking interference from medium degradation (e.g., agar melting at high temperatures affecting radial growth measurements).

Validity of the findings

1、Low Statistical Power: The reported power of 0.72 (line 155) is below the standard 0.8, increasing the risk of Type II errors (30% chance of missing true DEGs), especially problematic for the underpowered HT group.
2、The conclusion that "H₂O₂ decrease at high temperatures results from enhanced antioxidant enzymes" (lines 182-196) ignores the possibility of ROS scavenger overload due to cellular damage. This claim is unsupported by MDA-H₂O₂ correlation analysis.
3、The discussion (lines 441-447) assumes "ribosome gene upregulation in EHT indicates enhanced protein synthesis" but lacks Western blot/mass spectrometry data to confirm protein-level changes, ignoring post-transcriptional regulation.
4、Figure 2B shows increasing SOD activity with temperature, yet line 311 reports "CAT and SOD 2 genes were downregulated in the peroxisome pathway." This contradiction—active enzymes with downregulated genes—requires explanation (e.g., isoenzymes, post-translational activation), which is absent.
5、EHT-induced downregulation of glycolytic genes (Figure 8) is not linked to mycelial fresh weight reduction (Figure 1C) due to missing ATP content data, breaking the mechanistic link between energy metabolism and growth inhibition.
6、 "Abnormal morphology" under EHT (line 174) is not paired with microscopic structure analysis , leaving a gap between molecular data and phenotypic observations.
7、Missing Temporal Resolution: The study fails to distinguish short-term stress responses (e.g., rapid HSP induction) from long-term adaptations (gene expression remodeling). For example, Hsp20 upregulation in HT (lines 328-329) is not validated via time-course data to confirm it as a long-term adaptation.

Additional comments

Logical Rigor: Clarify short-term vs. long-term responses, avoid overgeneralization, and explicitly acknowledge limitations (single strain, short treatment)

---

## Round 0.2 · Minor Revisions

One Reviewer still has concerns about the temperature and nomenclature used. Please address this issue.

Reviewer 1 ·

Basic reporting

-

Experimental design

A: We sincerely appreciate your insightful suggestion regarding the nomenclature of the temperature conditions and the optimal growth environment for Auricularia heimuer. After careful consideration, we believe that the nomenclature should be based on the scientific literature and the specific growth characteristics of the organism. Previous studies indicate that the optimal growth temperature for Lentinula edodes mycelium is 24-27 ℃ (Guo et al., 2023) while the optimal growth temperature for Auricularia heimuer mycelium ranges from 25-30 ℃ (Jo et al., 2014). In our experiment, we selected 25 ℃ as the normal temperature (NT) condition, which aligns with the established optimal growth temperature range for this fungus. While the mycelia did exhibit vigorous growth at higher temperatures (35 ℃), it is important to note that these higher temperatures may still represent a form of stress for the organism. The term "high temperature (HT)" in our study refers to conditions that are above the optimal growth temperature but do not necessarily inhibit growth entirely.
Response:
According to the definition of optimal growth temperature, an organism shows the highest growth rate at its optimal temperature and the optimum growth temperatures can vary among different strains of the same organisms. Based on your results, your Auricularia heimuer strain clearly shows an optimal temperature at a higher level (35 °C) than the ones you mentioned, indicating that your strain is well adapted to high-temperature conditions, including thermal stress. This distinction should be clearly stated in your article to avoid confusion between normal temperature (NT) and optimal temperature in your research. Furthermore, it is important to discuss in more detail how your A. heimuer strain showed superior growth under high-temperature (HT) conditions.

Validity of the findings

-

Additional comments

-

Reviewer 2 ·

Basic reporting

no comment

Experimental design

no comment

Validity of the findings

no comment

Additional comments

no comment

---

## Round 0.3 · accepted · Accept

The authors properly addressed the Reviewers' comments. Consequently, the manuscript is suitable for the next editorial step.